# Spanish Adaptation and Validation of the Child Adjustment and Parent Efficacy Scale

**Dolores Seijo [1], David Tomé [2] , Jessica Sanmarco [3] , Alina Morawska [4] and Francisca Fariña [5,\*]**

1   School of Psychology, University of Santiago de Compostela, 15705 Santiago de Compostela, Spain;
    mariadolores.seijo@usc.es
2   Research Group PS1, University of Vigo, 36310 Vigo, Spain; dtomelourido@hotmail.com
3   Research Group 1754, University of Santiago de Compostela, 15705 Santiago de Compostela, Spain;
    Jessica.sanmarco.vazquez@usc.es
4   School of Psychology, University of Queensland, St Lucia, QLD 4072, Australia; alina@psy.uq.edu.au
5   AIPSE, University of Vigo, 36310 Vigo, Spain
*   Correspondence: francisca@uvigo.es

**Abstract:** (1) Background: The aim of this study is to obtain an instrument, with robust psychometric properties validated with a Spanish sample, that allows for the evaluation of the emotional and behavioural adjustment of children, as well as the perceived ability of parents to manage their children's problems. (2) Methods: Data from 2618 Spanish parents of 2–12-year-old children were used to evaluate the psychometric properties of the Child Adjustment and Parent Efficacy Scale. In order to develop the intensity scale, exploratory and confirmatory analyses were carried out, and the reliability, validity, and invariance of the measurement model were estimated. (3) Results: A structure of 25 items grouped into two factors are shown, which allows for the evaluation of emotional and behavioural problems and children's competencies. The model adjustment indicators were satisfactory. Favourable evidence was obtained for the reliability of the measurement model from two perspectives, internal consistency and composite reliability. The discriminant validity was satisfactory, as was the homogeneity of the measurement model based on child gender. Regarding the self-efficacy scale, confirmatory analysis procedures were also carried out, verifying a good factorial structure. (4) Conclusions: Results support a scale with robust psychometric properties that measure child adjustment and parent self-efficacy. The instrument can help to improve family effectiveness and be useful for schools and teachers to promote the well-being of children. The instrument could also be good to evaluate in special contexts, like custody evaluations in a legal or forensic context.

**Keywords:** adaptation; adjustment; emotional; children; parents; CAPES; educational quality

## 1. Introduction

In recent decades, scientific research has increased understanding of which factors improve psychological adjustment in children and which increase the risk for behavioural and emotional problems [1–3]. It is very important to obtain efficient measures of the emotional and behavioural adjustment of children, which allow for quality indicators in a short space of time and enable evaluation of intervention outcomes. Barlow et al. [4] conducted a systematic review to analyse parental training programs that served to improve child emotional and behavioural adjustment. Among the papers evaluated, the following instruments were used to measure children's behavioural, emotional, and social problems: Behavioural Screening Questionnaire [5]; Eyberg Child Behaviour Inventory [6]; Dyadic Parent-Child Interaction Coding System [7]; Strengths and Difficulties Questionnaire [8,9]; Child Behaviour Checklist [10]; Children's Behaviour Questionnaire [11]; and Behavioural Inhibition Questionnaire [12]. However, despite the existence of several widely evaluated instruments noted above, these have not been evaluated in a Spanish context. Only the Child Behaviour Checklist [13] has been used in Spain. However, in this case, the

adaptation was not done empirically but through cognitive interviews. Within the Spanish context, the Sistema de Evaluación de Niños y Adolescentes (SENA) (System for the Evaluation of Children and Adolescents) [14] is a measure that evaluates internalising (predominantly emotional manifestations), externalising (manifested as behavioural or disruptive problems), or contextual problems. SENA evaluates a broad spectrum of emotional and behavioural problems, contextual problems, areas of vulnerability, and psychological resources.

The original Child Adjustment and Parent Efficacy Scale (CAPES) [15] was created in English as a short instrument that presents two scales: the intensity scale measures child behavioural and emotional problems and the self-efficacy scale measures parental confidence to deal with these problems [16]. CAPES has obtained good psychometric data, has been translated and evaluated in other languages (e.g., Spanish–Panama [16], Chinese [17]), and has been adapted for children who have some type of disability [18]. The CAPES adaptation to Spanish was carried out in a sample of 174 Panamanian parents; however, these authors indicated that more research was needed on the psychometric properties of the instrument in its Spanish version before it could be widely used in practice. This article analyses the psychometric properties in relation to the intensity and self-efficacy scales with the objective of obtaining an instrument that allows for the evaluation of the emotional and behavioural adjustment of children in Spain, as well as the perceived ability of parents to manage their children's problems. CAPES can also be useful for families, schools, and teachers to promote the well-being of children.

## 2. Materials and Methods

### 2.1. Participants

The sample was composed of 2618 Spanish parents. Table 1 provides data on sociodemographic characteristics of this study.

**Table 1.** Socio-demographic characteristics.

| Variable | | N (%) |
|---|---|---|
| Age child | $M = 7.13$ ($SD = 2.73$). Range 2–12 | |
| Sex child | Male | 1311 (50.1%) |
| | Female | 1289 (49.2%) |
| Relationship to child | Father or foster father | 459 (17.5%) |
| | Mother or foster mother | 2127 (81.2%) |
| | Stepfather | 2 (0.1%) |
| | Stepmother | 19 (0.7%) |
| | Other | 5 (0.2%) |
| Marital status | Married | 1934 (73.9%) |
| | Cohabiting | 353 (13.5%) |
| | Divorced | 222 (8.5%) |
| | Single | 84 (3.2%) |
| | Widow | 9 (0.3%) |
| | Other | 9 (0.3%) |
| Family status | Both biological or adoptive parents | 2257 (86.2%) |
| | One biological or adoptive parent and one stepparent | 91 (3.5%) |
| | Single parent family | 103 (3.9%) |
| | Other | 129 (4.9%) |
| Children of previous relationships | Yes | 112 (4.3%) |
| | No | 2294 (87.6%) |

Note: Percentages may not add up to 100 due to missing data.

### 2.2. Measures

The Family Background Questionnaire (FBQ) [19] was used to measure socio-demographic data. The questionnaire selected for adaptation was the 30-item Child Adjustment and

Parental Efficacy Scale (CAPES) [15]. The original CAPES consists of two subscales, the intensity scale and the self-efficacy scale. The final version of the intensity scale is composed of 24 items that assess behaviour problems (e.g., "My child follows rules and limits") and the emotional maladjustment scale is composed of three items that assess emotional adjustment (e.g., "My child misbehaves at mealtimes") and three non-loading items. Parents rate each item from 0 (not true at all) to 3 (true most of the time) depending on how true the statement was for their child in the past 4 weeks. Items are summed to yield a total intensity score (CAPES intensity scale) composed of a behaviour score and an emotional maladjustment score. Higher scores indicate higher levels of problems. An additional self-efficacy scale asks parents to rate their confidence in handling the 20 problematic behaviour items [15]. Parents rate each item from 1 ("certain I can't do it") to 10 ("certain I can do it") depending on how confident they are in successfully dealing with their child's behaviour. Items are summed to yield a total self-efficacy score, with higher scores indicating higher levels of parental self-efficacy. The reliability of the original intensity scale was positive, with a high internal consistency global value, $\alpha = 0.90$, as well as a good value for the "behaviour" factor, $\alpha = 0.96$, and a moderate value for the "emotional maladjustment factor, $\alpha = 0.74$. Model fit based on confirmatory factor analysis was satisfactory $\chi^2(86) = 174.88$, $p < 0.001$; comparative fit index (CFI) = 0.952; standardized root mean square residual (SRMR) = 0.052; root mean square error of approximation (RMSEA) = 0.055 (90% CI; 0.043–0.066). Regarding the original self-efficacy scale, the reliability was good, $\alpha = 0.96$, as well as the model fit.

*2.3. Procedure*

Data for this study were collected as part of the International Parenting Survey (IPS) [20] in Spain. IPS is developed in primary schools. In order to adapt the questionnaire to the Spanish context, the original version created by Morawska et al. [15] was adapted through a translation/back translation method [21]. Regarding data collection, first of all, permission to carry out the research was obtained by the bioethics committee of the University of Santiago de Compostela (Spain). An accidental sampling was performed, without estimates of sample size, although seeking to reach a minimum of 400 participants according to the sampling formula for infinite populations. The objective was that as many parents as possible who met the inclusion criteria participated, so that the final sample was diverse and representative of the Spanish population. Next, we contacted numerous schools from different geographical locations and socioeconomic environments. School managers sent letters to parents inviting them to participate in the study (6277 parents with 41.71% response rate). Parents were informed about the research objectives and the anonymous and voluntary nature of the data collection. Instructions for completing the questionnaire were the same for all parents, and they were written in the document that they had to answer. Parents who chose to participate returned completed questionnaires as well as an informed consent document. All the questionnaires were hard copy. At all times, the data were processed according to Spanish data protection laws. As an inclusion criterion, it was established that the parents should have a child aged 2 through 12.

*2.4. Data Analysis*

The data analyses were carried out sequentially through the IBM SPSS version 25 and IBM AMOS version 25 statistical packages following the data analysis design of Fariña, Arce, Tomé, and Seijo [22] for the adaptation and validation of a psychometric instrument. Regarding the CAPES intensity scale, first, the descriptive statistics of the 30 items adapted from the original model were calculated. Subsequently, the sample was divided into two groups of participants (the first 1309 and the last 1309 parents). Since the relationship between items and factors was different in the original model [15] and in the first adaptation to Panama [16], an exploratory factor analysis was performed with one of the parts (calibration sample), after which 25 items were selected for the Spanish version. Next, with the other part of the sample (validation sample), confirmatory factor analysis

procedures were used in order to obtain additional criteria for model adjustment, as exploratory methods may be valuable to focus hypotheses for the confirmatory analyses so they can be used in a complementary way [23]. With the data obtained in the confirmatory factor analysis, evidence was obtained about the reliability and validity of the factors. The invariance of the measurement model was calculated based on child gender. Finally, and considering the entire sample, the factor structure of the CAPES self-efficacy scale was tested, based on the factor model previously found.

## 3. Results

### 3.1. Descriptive Statistics

Table 2 shows minimum and maximum values obtained in the 30 initial items of the questionnaire, as well as statistics for mean (*M*), standard deviation (*SD*), skewness, and kurtosis. The averages of the items are between 0.64 (item 27) and 2.73 (item 12), while typical deviations have values below 1.08. Skewness values were negative for the 30 items. Considering kurtosis, it is positive in 21 items and negative in 9. For making factor models, a multivariate normal distribution is required, so skewness scores in the range of $\pm3$, those of kurtosis in $\pm10$ [24], and all scores were within this range.

**Table 2.** CAPES intensity: descriptive statistics of the items.

| Items | *M* | *SD* | Skewness *SE* = 0.048 | Kurtosis *SE* = 0.096 |
|---|---|---|---|---|
| CAPES1. Se altera o enfada cuando no se sale con la suya. | 1.56 | 0.80 | −0.36 | −0.33 |
| CAPES2. Se niega a realizar tareas del hogar cuando se le pide. | 2.10 | 0.73 | −0.64 | 0.46 |
| CAPES3. Se preocupa. | 1.54 | 0.90 | −0.13 | −0.73 |
| CAPES4. Le dan pataletas. | 2.13 | 0.81 | −0.64 | −0.18 |
| CAPES5. Se porta mal durante las comidas. | 2.27 | 0.76 | −0.96 | 0.73 |
| CAPES6. Discute o se pelea con otros niños/as o con sus hermanos/as. | 2.00 | 0.77 | −0.54 | 0.13 |
| CAPES7. Rechaza comerse la comida. | 2.11 | 0.80 | −0.73 | 0.26 |
| CAPES8. Tarda mucho en vestirse. | 1.97 | 0.86 | −0.54 | −0.33 |
| CAPES9. Me lastima a mi o a otros (golpea, araña, muerde, empuja . . . ). | 2.69 | 0.58 | −2.02 | 4.18 |
| CAPES10. Interrumpe cuando hablo con otras personas. | 1.74 | 0.76 | −0.35 | −0.05 |
| CAPES11. Parece asustado o temeroso. | 2.49 | 0.68 | −1.34 | 1.66 |
| CAPES12. Tiene problemas de comportamiento en la guardería o el colegio. | 2.73 | 0.56 | −2.40 | 6.38 |
| CAPES13. Tiene dificultades para entretenerse sin la atención de un adulto. | 2.53 | 0.72 | −1.55 | 1.98 |
| CAPES14. Pega gritos, chillidos o es escandaloso. | 2.27 | 0.78 | −0.91 | 0.33 |
| CAPES15. Lloriquea o se queja. | 2.13 | 0.75 | −0.68 | 0.33 |
| CAPES16. Muestra una actitud desafiante cuando se le pide algo. | 2.44 | 0.68 | −1.15 | 1.20 |
| CAPES17. Llora más que otros/as niños/as de su edad. | 2.66 | 0.65 | −2.14 | 4.51 |
| CAPES18. Me contesta de forma grosera. | 2.58 | 0.63 | −1.54 | 2.45 |
| CAPES19. Aparenta estar descontento o triste. | 2.62 | 0.61 | −1.76 | 3.34 |
| CAPES20. Tiene dificultades para organizar las tareas y actividades. | 2.28 | 0.80 | −1.00 | 0.48 |
| CAPES21. Acepta las reglas y límites. | 1.82 | 1.01 | −0.26 | −1.13 |
| CAPES22. Se lleva bien con los miembros de la familia. | 2.34 | 1.08 | −1.39 | 0.33 |
| CAPES23. Es bondadoso/a y servicial con los demás. | 2.24 | 0.99 | −1.13 | 0.07 |
| CAPES24. Es capaz de entretenerse sin la constante atención adulta. | 2.09 | 1.02 | −0.84 | −0.48 |
| CAPES25. Coopera a la hora de dormirse. | 2.09 | 1.01 | −0.81 | −0.54 |
| CAPES26. Parece sentirse bien consigo mismo/a. | 2.28 | 0.98 | −1.26 | 0.41 |
| CAPES27. Se lleva bien con otros niños/as. | 2.36 | 0.97 | −1.46 | 0.94 |
| CAPES28. Expresa sus puntos de vista, ideas y necesidades de manera adecuada. | 2.15 | 0.95 | −0.94 | −0.10 |
| CAPES29. Puede realizar tareas pertinentes a su edad por sí mismo/a. | 2.35 | 0.97 | −1.40 | 0.75 |
| CAPES30. Obedece las instrucciones de los adultos. | 2.14 | 0.88 | −0.91 | 0.16 |

Note: All items had a minimum score of 0 and a maximum score of 3.

### 3.2. Exploratory Factor Analysis

The reduction of the dimensions was carried out using the "principal components" extraction method. The rotation method used was "direct oblimin with Kaiser normalization". The value of the sample adequacy measure, Kaiser–Meyer–Olkin, was satisfactory (KMO = 0.913), in the same way as Bartlett's sphericity test, $\chi^2(435) = 18{,}115.55$, $p < 0.001$. The rotation that converged in 33 iterations is presented in Table 3.

**Table 3.** Factorial matrix rotated.

| Items | Factor | | | | | |
|---|---|---|---|---|---|---|
| | **1** | **2** | **3** | **4** | **5** | **6** |
| CAPES27 | 0.864 | | | | | |
| CAPES26 | 0.849 | | | | | |
| CAPES29 | 0.836 | | | | | |
| CAPES23 | 0.830 | | | | | |
| CAPES22 | 0.806 | | | | | |
| CAPES30 | 0.805 | | | | | |
| CAPES24 | 0.793 | | | | | |
| CAPES28 | 0.789 | | | | | |
| CAPES25 | 0.762 | | | | | |
| CAPES21 | 0.599 | | | | | |
| CAPES15 | | 0.683 | | | | |
| CAPES14 | | 0.622 | | | | |
| CAPES16 | | 0.61 | | | | |
| CAPES1 | | 0.608 | | | | |
| CAPES4 | | 0.552 | −0.470 | | | |
| CAPES18 | | 0.52 | | −0.416 | | |
| CAPES10 | | 0.516 | | | | |
| CAPES2 | | 0.501 | | | | |
| CAPES5 | | 0.497 | | 0.435 | | |
| CAPES9 | | 0.494 | | | | |
| CAPES17 | | 0.481 | | | | 0.412 |
| CAPES13 | | 0.444 | | | −0.424 | |
| CAPES8 | | 0.419 | | | | |
| CAPES19 | | 0.448 | 0.545 | | | |
| CAPES20 | | 0.431 | 0.487 | | | |
| CAPES11 | | | 0.468 | | | |
| CAPES7 | | | | 0.640 | | |
| CAPES3 | | | | | 0.559 | |
| CAPES12 | | 0.451 | | | | −0.467 |
| CAPES6 | | | | | | |
| **Eigenvalues** | 7.027 | 5.341 | 1.602 | 1.337 | 1.183 | 1.01 |

Note: Items with factor loadings lower than the value 0.4 were suppressed, and were not considered to be part of any factor.

In order to determine the number of factors, the following criteria were considered: (1) the eigenvalues were greater than one; (2) factors had a minimum of three items; (3) a substantive interpretation based on the two factors present, both on the original scale [15] and on the adaptation to Panama [16].

The exploratory factor analysis showed two clear factors that explained much of the variance (exceeding the minimum threshold of 40%). In addition, these factors had more than three items and theoretically corresponded to the previous CAPES models [16–18]. Therefore, and verifying that the three previous criteria were met, the new model was composed of two factors that explain 41.23% of the variance. According to the factor names proposed by Mejia et al. (16), the first factor, "child's competencies", included items from 21 to 30 ($M = 2.17$, $SD = 0.82$), and the second factor, "behavioural and emotional problems", included items 1, 2, 4, 5, 8, 9, 10, 13, 14, 15, 16, 17, 18, 19, and 20 ($M = 2.28$, $SD = 0.40$). The rest of the factors did not meet the second and third criteria (they did not reach three items,

nor was there a theoretical basis to conform them). Although factors 3 and 4 could have three or more items, the theoretical content of these items corresponds to factors 1 and 2 (leaving, in this case, fewer than three items without a theoretical interpretation). Therefore, in subsequent analyses, items that were not part of these two factors were disregarded. Guidelines for the interpretation of the scores in forensic setting evaluation case studies [25] are provided in Appendix A.

### 3.3. Confirmatory Factor Analysis

The model shown in Figure 1 was specified according to the exploratory analysis relationship between items and factors.

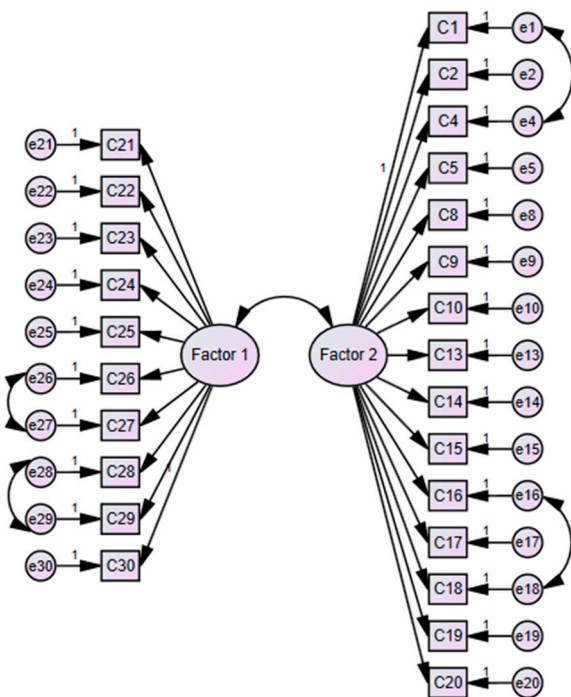

**Figure 1.** CAPES intensity model.

This model was over-identified with 325 elements in the variance-covariance matrix, with 51 parameters for estimation (25 factor loadings, 25 error variances, and 1 factor correlation) and 270 degrees of freedom.

Factor and error variance were constrained to 1, leaving 25 factor loadings and 25 correlations free in order to avoid under-identification. In order to maximise the model's overall goodness of fit to the data, the model was re-specified including the correlations between items 26 and 27, and between 28 and 29 of the child's competencies factor, as well as the correlations between items 1 and 4, and between 16 and 18 of the behavioural and emotional problems factor.

The parameters were estimated by the maximum likelihood method and bootstrap to estimate the standard errors. Table 4 shows the values of factor loadings ($\lambda$) and error variances ($\delta$). Statistical significance was attained for all parameters ($p < 0.001$). Corrected item-total correlations were greater than 0.50 for all items of the first factor, except item 21 (0.34), while all items of the second factor did not exceed that threshold. There is a slight correlation between the two factors of the questionnaire, r = 0.113 and $p < 0.001$.

The model's overall goodness of fit indices were satisfactory [26]: $\chi^2$ = 1623.18, $p < 0.001$; $\chi^2/\mathrm{df}$ = 6.01; goodness of fit index (GFI) = 0.906; Tucker–Lewis Index (TLI) = 0.894; CFI = 0.905; normed fit index (NFI) = 0.888; RMSEA = 0.062 (90% CI; 0.059–0.065); and SRMR = 0.0473.

**Table 4.** CAPES intensity. Standardized regression weights and error variances.

| Factor | Item | Standardized Regression Weights (λ) | Error Variances (δ) | Corrected Item-Total Correlation |
|---|---|---|---|---|
| 1. Child's Competencies | CAPES 21 | 0.58 | 0.66 | 0.56 |
| | CAPES 22 | 0.80 | 0.40 | 0.79 |
| | CAPES 23 | 0.81 | 0.33 | 0.81 |
| | CAPES 24 | 0.74 | 0.46 | 0.73 |
| | CAPES 25 | 0.74 | 0.46 | 0.72 |
| | CAPES 26 | 0.82 | 0.31 | 0.82 |
| | CAPES 27 | 0.86 | 0.22 | 0.86 |
| | CAPES 28 | 0.75 | 0.39 | 0.75 |
| | CAPES 29 | 0.83 | 0.29 | 0.82 |
| | CAPES 30 | 0.76 | 0.32 | 0.75 |
| 2. Behavioural and Emotional Problems | CAPES 1 | 0.58 | 0.44 | 0.55 |
| | CAPES 2 | 0.44 | 0.45 | 0.43 |
| | CAPES 4 | 0.57 | 0.45 | 0.52 |
| | CAPES 5 | 0.46 | 0.45 | 0.43 |
| | CAPES 8 | 0.44 | 0.64 | 0.4 |
| | CAPES 9 | 0.50 | 0.24 | 0.46 |
| | CAPES 10 | 0.52 | 0.44 | 0.45 |
| | CAPES 13 | 0.44 | 0.43 | 0.39 |
| | CAPES 14 | 0.64 | 0.38 | 0.58 |
| | CAPES 15 | 0.63 | 0.33 | 0.6 |
| | CAPES 16 | 0.58 | 0.31 | 0.55 |
| | CAPES 17 | 0.50 | 0.34 | 0.44 |
| | CAPES 18 | 0.50 | 0.31 | 0.49 |
| | CAPES 19 | 0.35 | 0.33 | 0.35 |
| | CAPES 20 | 0.48 | 0.55 | 0.42 |

*3.4. Reliability Analysis*

Factor reliability was calculated from two perspectives: Cronbach's alpha and composite reliability (see Table 5). In all cases, the values were satisfactory [27,28]. Therefore, on the one hand, considering the perspective of internal consistency measured through Crobach's alpha, the values indicate that the items that make up each of the two factors provide similar results and are consistent for measuring the constructs in the population. On the other hand, the composite reliability values show that the amount of true score variance relative to the total scale score variance is high, being a robust indicator of the usefulness of the items to evaluate the composite constructs.

**Table 5.** CAPES intensity: reliability index.

| Factor | Cronbach's Alpha (All Sample) | Cronbach's Alpha (EFA Sample) | Cronbach's Alpha (CFA Sample) | Composite Reliability (CFA Sample) |
|---|---|---|---|---|
| 1. Child's competencies | 0.94 | 0.95 | 0.93 | 0.94 |
| 2. Behavioural and emotional problems | 0.84 | 0.84 | 0.84 | 0.91 |

*3.5. Convergent and Discriminant Validity*

In order to determine the convergent validity of each of the two factors of the questionnaire, to find out if they really measure a specific construct, the average variance extracted (AVE) was used. In the case of the child´s competencies factor, the AVE was 0.61, while the AVE of the behavioural and emotional problems factor had a value of 0.39. With regard to discriminant validity, the discriminant validity of the AVE of each of the factors of the questionnaire is greater than the square of the correlations between them (0.012).

### 3.6. Invariance of the Measurement Model

Finally, an invariance analysis was performed to determine if the factor model was homogeneous based on various values of a multi-group moderator. The "sex of the child" variable was chosen to divide the sample into two groups, on the one hand the parents of boys (N = 1311), and on the other, parents of girls (N = 1289). To determine if the factor structure was invariant, and following the criteria of Cheung and Rensvold [29], the CFI differences between an unconstrained model (CFI = 0.907) and a constrained model (CFI = 0.907) were checked. There was no difference between both models. In addition, the freely estimated model (unconstrained) for the two groups achieved a good model fit: $\chi^2/df = 6.20$; GFI = 0.903; TLI = 0.897; NFI = 0.892; and RMSEA = 0.045 (90% CI; 0.043–0.046).

### 3.7. CAPES Self-Efficacy Scale

A factorial model was specified based on the perceived self-efficacy of mothers and fathers regarding the emotional and behavioural problems of their children. The factor specification, based on the previous factor structure of the "behavioural and emotional problems" factor, can be found in Figure 2.

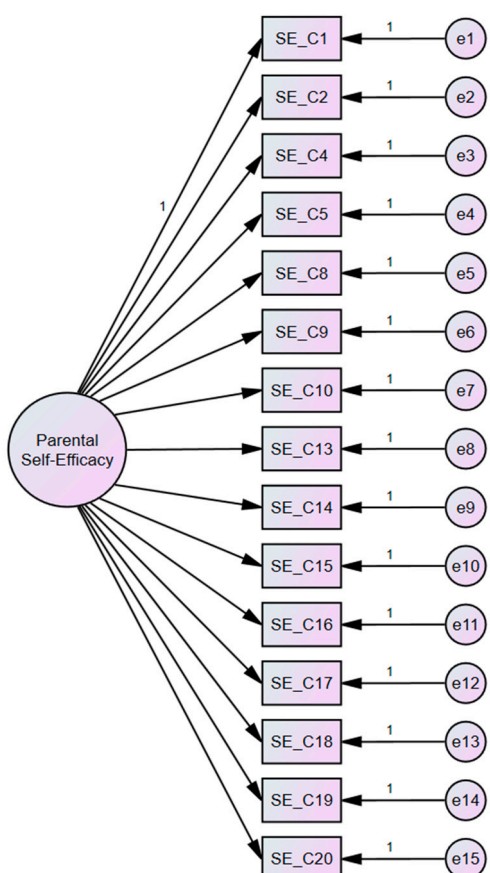

**Figure 2.** CAPES self-efficacy model.

Due to missing data, the parameters were calculated using the maximum likelihood method, estimating means and intercepts. As in the previous factor analysis, the variances of the factors and errors were constrained to 1 to avoid under-identification of the model. Therefore, the model was over-identified with 135 elements in the variance-covariance matrix: 47 parameters to estimate and 88 degrees of freedom. In order to achieve a better fit, the model was re-specified, allowing correlations between items SE_C1 and SE_C4, as well as between items SE_C19 and SE_CE20. These re-specifications were also made in the model created by Mejia et al. [16].

The goodness indices of the model global adjustment, which could be calculated due to the lost values, were acceptable: $\chi^2$ = 1307.782, $p < 0.001$; $\chi^2/\mathrm{df}$ = 14.86; TLI = 0.910; CFI = 0.934; and RMSEA = 0.073 [IC 90% 0.069, 0.076].

The parameters to be estimated in the model reached statistical significance ($p < 0.001$), while the corrected item-total correlations were greater than 0.5 for all items. Table 6 shows the factor loads (all of them greater than 0.7) and the errors variances of the items.

**Table 6.** CAPES self-efficacy. Standardised regression weights and error variances.

| Items | Standardised Regression Weights (Λ) | Error Variances (δ) | Corrected Item-Total Correlation |
|---|---|---|---|
| SE_C1 | 0.758 | 0.425 | 0.69 |
| SE_C2 | 0.768 | 0.41 | 0.71 |
| SE_C4 | 0.776 | 0.398 | 0.71 |
| SE_C5 | 0.78 | 0.392 | 0.71 |
| SE_C8 | 0.748 | 0.44 | 0.71 |
| SE_C9 | 0.847 | 0.283 | 0.79 |
| SE_C10 | 0.759 | 0.424 | 0.71 |
| SE_C13 | 0.853 | 0.272 | 0.81 |
| SE_C14 | 0.868 | 0.247 | 0.84 |
| SE_C15 | 0.902 | 0.186 | 0.84 |
| SE_C16 | 0.908 | 0.176 | 0.83 |
| SE_C17 | 0.901 | 0.188 | 0.83 |
| SE_C18 | 0.893 | 0.203 | 0.81 |
| SE_C19 | 0.862 | 0.257 | 0.72 |
| SE_C20 | 0.776 | 0.398 | 0.71 |

The factor composed of these 15 items reached a Cronbach's alpha of 0.97, which did not improve by eliminating any of the items. Similarly, the composite reliability data of the factor was also 0.97. Finally, Table 7 shows the descriptive statistics for the items and for the factor.

**Table 7.** CAPES self-efficacy: descriptive statistics of the items.

| Items | Minimum | Maximum | *M* | *SD* | Skewness | Skewness *SE* | Kurtosis | Kurtosis *SE* |
|---|---|---|---|---|---|---|---|---|
| SE_C1 | 0 | 10 | 7.84 | 2.16 | −1.29 | 0.06 | 1.60 | 0.13 |
| SE_C2 | 0 | 10 | 8.27 | 2.06 | −1.85 | 0.06 | 3.79 | 0.13 |
| SE_C4 | 0 | 10 | 7.91 | 2.39 | −1.54 | 0.07 | 2.08 | 0.13 |
| SE_C5 | 0 | 10 | 7.94 | 2.44 | −1.60 | 0.07 | 2.28 | 0.13 |
| SE_C8 | 0 | 10 | 7.89 | 2.25 | −1.43 | 0.07 | 2.09 | 0.14 |
| SE_C9 | 0 | 10 | 8.47 | 2.50 | −2.14 | 0.07 | 4.01 | 0.14 |
| SE_C10 | 0 | 10 | 7.68 | 2.22 | −1.32 | 0.07 | 1.74 | 0.14 |
| SE_C13 | 0 | 10 | 8.24 | 2.33 | −1.89 | 0.07 | 3.57 | 0.14 |
| SE_C14 | 0 | 10 | 7.93 | 2.35 | −1.56 | 0.07 | 2.26 | 0.14 |
| SE_C15 | 0 | 10 | 7.92 | 2.25 | −1.64 | 0.07 | 2.79 | 0.14 |
| SE_C16 | 0 | 10 | 8.18 | 2.32 | −1.92 | 0.07 | 3.75 | 0.14 |
| SE_C17 | 0 | 10 | 8.25 | 2.48 | −1.94 | 0.07 | 3.42 | 0.14 |
| SE_C18 | 0 | 10 | 8.32 | 2.47 | −2.04 | 0.07 | 3.78 | 0.14 |
| SE_C19 | 0 | 10 | 8.25 | 2.46 | −2.01 | 0.07 | 3.67 | 0.14 |
| SE_C20 | 0 | 10 | 7.91 | 2.53 | −1.66 | 0.07 | 2.33 | 0.14 |
| TOTAL | 0 | 10 | 8.13 | 1.97 | −2.17 | 0.08 | 5.36 | 0.15 |

## 4. Discussion

In this study, the Child Adjustment and Parent Efficacy Scale was validated in the Spanish context. Although there was already a previous version of the items in Spanish [16], adapting the items to the Spanish context allows for greater validity in the use of the questionnaire [29].

Regarding the intensity scale, the results obtained reflect a factorial structure with two factors, which measure, on the one hand, the abilities of children (child competencies), and, on the other, their possible maladjustment (behavioural and emotional problems). This model of the intensity scale and all the items are similar and consistent, across the two factors, to those obtained by Mejia et al. [16] and field literature [30], as the items that configured the child competencies factor are included in the same factor in this adaptation, but the behavioural and emotional factor is only made up of 15 items in this case, and different from that created by Morawska et al. [15], where the items were grouped into two types of problems (emotional or behavioural) and there was no specific factor for children's competencies.

Concerning the adjustment of the structure of the measurement model, the indicators of both exploratory and confirmatory analysis refer to a satisfactory adjustment of the data, globally and in each of the parameters [31,32]. The data related to the reliability of the questionnaire were also satisfactory for both factors and for the entire instrument, both from the perspective of internal consistency with Cronbach's alpha measurement [33], and for composite reliability [26,27], in both cases exceeding the threshold of 0.7. Regarding the indicators of the validity of the measurement model, the data report a good convergent validity for the child competencies factor when exceeding the threshold of 0.5 [34], but not for the factor from behavioural and emotional problems. This second factor does not share more than half of the variance with its indicators, the majority being part of the measurement error. Regarding the discriminant validity, the VME of each construct is greater than the square of the correlations between the latent variables of the model, so that each of the factors measures a different construct [35]. The last of the intensity scale data analyses carried out shows that there is invariance in the measurement model between the two groups created according to child sex. Following the criteria of Cheung and Rensvold [29], the difference in CFI values between the unconstrained and constrained measurement models did not exceed the threshold of 0.01. It is assumed that the factor structure is homogeneous for the entire sample. The fact that the unconstrained model for the two groups achieved a reasonable model fit implies good configurational invariance and the equivalence of the groups in relation to the factorial structure. Finally, the analyses performed with the parental self-efficacy scale reflect a good fit, as well as satisfactory reliability data [32,34]. Therefore, the scale that reflects the factor structure of the intensity scale factor "behavioural and emotional problems" is consistent in knowing how parents rate their ability to cope with these problems [35,36].

The limitations of the study (measure instrument) are related to the sources of the respondent bias [37], mainly item social desirability in a forensic setting assessment. As for controlling it, the instrument should be administered in this setting together with a measure of defensiveness [38].

The results support a scale with strong psychometric properties that measures the adjustment of the child and the self-efficacy of the parents. Specifically, CAPES is an instrument with two factors that assess children's skills and competencies and behavioural and emotional problems. Additionally, it adds a scale that provides an indicator of parental self-efficacy.

CAPES could be useful for professionals involved with parents in order to improve the positive exercise of parenting, or for those who have a high conflict [39] either in the educational or school context [40,41], or in a legal setting, for example in the judicial assessment of custody [42], in family mediation [43], in parenting coordination [44], in family court therapy [45], and in any other type of intervention in which the inter-parent relationship has to be improved. Future research could specifically study these applications.

**Author Contributions:** The individual contribution of the authors are: conceptualization, D.S. and F.F.; methodology, D.T., J.S. and F.F.; validation, D.T., J.S. and F.F.; formal Analysis, D.T., J.S. and F.F.; investigation, A.M., D.S. and F.F.; resources, D.S. and F.F.; writing—original draft preparation, D.S., J.S., A.M. and F.F.; writing—review & editing, D.S., A.M. and F.F.; supervision, F.F.; funding acquisition, D.S. and F.F. All authors have read and agreed to the published version of the manuscript.

**Funding:** This research was funded by a grant of the Consellería de Cultura, Educación e Ordenación Universitaria of the Xunta de Galicia (ED431B 2020/46) and by a grant of the Spanish Ministry of Economy and Competitiveness (PSI2017-87278-R).

**Institutional Review Board Statement:** Not Applicable.

**Informed Consent Statement:** Not Applicable.

**Data Availability Statement:** Not Applicable.

**Conflicts of Interest:** The authors declare no conflict of interest.

### Appendix A

**Table A1.** Score interpretation table.

| Raw Score | Cut Score | Interpretation |
|---|---|---|
| **Child competence subscale [a]** | | |
| $\leq 8.37$ | $\leq -1.645$ | Significant incompetence |
| $>8.37$ and $<11.39$ | $\leq -1.28$ and $>-1.645$ | Moderate incompetence |
| $>32.53$ and $<35.55$ | $\geq 1.28$ and $<1.645$ | Moderately high competence |
| $\geq 35.55$ | $\geq 1.645$ | Significantly high competence |
| **Behavioural and emotional problems [b]** | | |
| $\leq 0.83$ | $\leq -1.645$ | Significantly high no-problems [c] |
| $>0.83$ and $<3.08$ | $\leq -1.28$ and $>-1.645$ | Moderately high no-problems [c] |
| $>18.90$ and $<21.16$ | $\geq 1.28$ and $<1.645$ | Moderate deteriorate |
| $\geq 21.16$ | $\geq 1.645$ | Significant deteriorate |

Note: [a] asymmetry = 1.44 and kurtosis = 1.05; [b] asymmetry = 0.75 and kurtosis = 0.86; [c] in forensic evaluation suspect faking good response bias.

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
