# Peer review of "Spanish Adaptation and Validation of the Child Adjustment and Parent Efficacy Scale"

_sustainability, doi:10.3390/su13094647_

Round 1

Reviewer 1 Report

I consider the article to be very well organized and of great interest to readers. 
I only consider that the data tables are very extensive; there could be a balance between the description of the data and their presentation. 

Reviewer 2 Report

Thank you for the opportunity to review the interesting manuscript entitled "Spanish adaptation and validation of the Child Adjustment and Parent Efficacy Scale". The sample chosen is robust, and the analyses were well executed and fitted the research questions. I enjoyed reading the manuscript. Please see my suggestions below.

  • In the method section, the authors state that inclusion criteria concerned having a child between 2 - 12 years of age. How then can the age range (as indicated in the Table in the participants section) include children of 1 year old?
  • The authors state that the CAPES was adapted to fit the Spanish context. I assume they mean that the questionnaire was adapted to the context in Spain. Use of the word "Spanish" here is slightly confusing (as the Panamese version was also in Spanish). Please clarify all instances of the use of "Spanish" when it is used to defer to "the context in Spain" rather than the language. 
  • Although the written language is clear, I did identify quite a number of language errors. Please proofread and correct prior to resubmission. A number of examples include:
    • L.34 "the child..."
    • L.48 their
    • L.44 relatively sample
    • L.58We advance research
    • L.61 Also can result
    • L.62 promove
    • L.102 send
    • L.159-161 - in general these two lines were not clear to me.

I wish the authors well in revising their manuscript!

Reviewer 3 Report

This study was focussed on the statistical assessment of characteristics the Child Adjustment and Parent Efficacy Scale that was translated into Spanish. Analysis of the problem was based on a sample of 2618 parents who were surveyed using hard questionnaires. Standard statistical and psychometric characteristics were assessed to analyse validity of the instrument. The presented results show that the authors are positive about the validity of this instrument and recommend its use in research.

Reading the submitted manuscript, several questions arose and inaccuracies were noticed, which I recommend to fix.

  1. Abstract: The aim and relevant conclusions (not just practical implications) must be provided. Keywords may include "Parents".
  2. Introduction: It is adequate. I recommend indicating in the text the authors of CAPES and in which language the instrument was created.
  3. Materials and Methods: A few words about sample size estimation and response rate would be needed here. Why so large sample (n=2618) was needed, as for validation studies a smaller sample is appropriate? There is a confusion with the number of items (line 72): 30=24+6+19? Statistical analysis must be included as a separate section.
  4. Results: There are a lot of technical details but a few explanation of the meaning of presented estimations. Table 3, indicate that small coefficients with absolute value below 0.40 are suppressed. Tables 4 and 6, indicate that standardized estimations, an additional column with item-total r is needed. Table 5 needs more comments, what does role play composite reliability? Line 186, [citation]. Where is Appendix 2? Appendix A is empty.
  5. Discussion: Line 245, "with ten items each"? Are there any limitations of the study?
  6. Conclusions: The presented text describes implications of the study, but not conclusions.

Thank you for considering my opinion. I encourage authors to keep on working to improve the manuscript.

Round 2

Reviewer 3 Report

Dear Authors,

I consider that the most of changes introduced by the authors are adequate to clarify the aspects that were pointed out in the review, so compared to the previous version, the article is improved. However, there are some issues left that I recommend fix:

  1. Overall:

1.1. It was still unclear what parts the validated scale consisted of. The Measures section states that "The questionnaire selected for adaptation was the 30-item Child Adjustment and Parental Efficacy Scale (CAPES) [15]. The original CAPES consists of two subscales, the Behavior and Emotional Maladjustment Scale and the Self-Efficacy Scale. The Behavioral Scale is composed of 24 items that assess behavioral problems (eg, My child follows rules and limits) and the Emotional Maladjustment Scale is composed of 3 items that assess emotional adjustment (eg, My child misbehaves at mealtimes) and 3 non-loading items." So this CAPES contains 30 original items. Next "An additional Self-Efficacy Scale asks parents to rate their confidence in handling the 19 problematic behavior items [15]." Why only 19 items here if The Behavioral Scale is composed of 24 items?

1.2. What is the relationship between the numbering of the items listed in Tables 2, 3, 4, Figure 1, and Appendix 1?

1.3. How could an EFA be done with 30 items (Table 2) if there were only 25 questions in the questionnaire (Appendix 1)?

1.4. It is necessary that Appendix 1 cover all the issues discussed in this article.

  1. The Introduction and Discussion should provide facts to present the facts why the Parent Self-Efficacy Scale is needed. Can any conclusions be drawn about CAPES from the results of this scale study?
  2. It is recommended to provide a list of abbreviations. In the text, each abbreviation must be preceded by a full explanation (see, for example, L49, L280).
  3. Table 1: Indicate total N and do space between N and (%).
  4. L49-51. What is the meaning of this sentence?
  5. L88-95. Are these data results of the present study or literature data? If the last, reference is needed.
  6. Section 2.3. Procedure: The is a mix of ideas. I recommend rewriting this section consistently.
  7. L170-173: This text is an issue of the Methods section.
  8. Appendix 2: How to evaluate these values: Child competence subscale: ³11.39 and £32.53; Behavioral and emotional problems subscale: ³3.08 and £18.90? Spelling: 'sore', 'faking'. In general, who needs this supplement?
  9. Reference #25 is wrong (instead 'harm' must be 'injury'). This reference needs to be removed.

Thank you for considering my opinion. I encourage authors to keep on working to improve the manuscript.

Author Response

First of all, we would like to thank you for the good recommendations you have made to improve the work. In the attachment we have given answer. We are waiting for your considerations.

Thank you

Best regards

Round 3

Reviewer 3 Report

The authors responded adequately to my comments and made appropriate corrections to the manuscript.